

# 1  Are cities responsible for their air
# 2  pollution?

Philippe Thunis[1], Alain Clappier[2], Alexander de Meij[3], Enrico Pisoni[1], Bertrand Bessagnet[1],
Leonor Tarrason[4].
[1] European Commission, Joint Research Centre, Ispra, Italy
[2] Université de Strasbourg, Laboratoire Image Ville Environnement, Strasbourg, France
[3] MetClim, Varese, Italy
[4] NILU, Norway
*Correspondence to*: Philippe Thunis (philippe.thunis@ec.europa.eu)

## 13  Abstract

While the burden caused by air pollution in urban areas is well documented, the origin of this
pollution and therefore the responsibility of the urban areas in generating this pollution is still a
subject of scientific discussion. Source Apportionment represents a useful technique to quantify
the city responsibility but the approaches and applications are not harmonized, therefore not
comparable, resulting in confusing and sometimes contradicting interpretations. In this work, we
analyze how different source apportionment approaches apply to the urban scale and how their
building elements and parameters are defined and set. We discuss in particular the options
available in terms of indicator, receptor, source and methodology. We show that different
choices for these options lead to very large differences in terms of outcome. In average over the
150 EU large cities selected in our study, the choices made for the indicator, the receptor and the
source each lead to an average factor 2 difference. We also show that temporal and spatial
averaging processes applied to the air quality indicator, especially when diverging source
apportionments are aggregated into a single number lead to favor strategies that target
background sources while occulting actions that would be efficient at the city center. We stress
that methodological choices and assumptions most often lead to a systematic and important
underestimation of the city responsibility, with important implications. Indeed, if cities are seen
as a minor actor, plans will target in priority the background at the expense of potentially
effective local actions.
**Keywords**: air pollution, source apportionment, particulate matter

## 35  1. Introduction

About 55% of the world's population lives in urban areas nowadays, and this number is expected
to increase to 68% by 2050, according to the United Nations (UN 2018). Large population
growth is also projected by 2030 in most of the major European cities (Alberti et al., 2019) with
predicted population growth varying in range from Berlin (15%), Paris (19%), Milan/Rome
(21%), Prague (37%), London (39%), to Brussels (52%) (see
https://urban.jrc.ec.europa.eu/thefutureofcities/urbanisation#the-chapter).  As a result of this



population trend, urban emissions and their associated pollution levels are expected to increase
as well.
According to a recent estimate (EEA, 2020), about 74 % of the EU-28 urban population are
exposed to pollution of fine particulate matter (PM$_{2.5}$) in concentrations above the WHO Air
Quality Guidelines value, this number raises to 99% for ozone (O$_3$) and is about 4% for nitrogen
dioxide (NO$_2$). Air pollution is a heavy burden on human health with more than 380,000
premature deaths in EU-28 reported in 2017 according to the same EEA estimates. For a wide
range of European cities, Khomenko et al. (2021) showed that the health burden due to air
pollution varies greatly by city, with annual premature mortality reaching up to 15% for PM$_{2.5}$
and 7% for NO$_2$. The highest mortality burden for PM$_{2.5}$ occurs in northern Italy, southern
Poland and eastern Czech Republic. De Bruyn and de Vries (2020) showed that for all 432 cities
in their sample (total population: 130 million inhabitants), the social costs (e.g. hospital
admissions, premature mortality) but also due to air pollution exceeded € 166 billion in 2018 for
Europe (EU27 plus the UK, Norway and Switzerland). City size was shown to be a key factor
contributing to the total social costs: all cities with a population over 1 million features in the
Top 25 cities with the highest social costs due to air pollution.
Given the health and economic burden caused by air pollution in urban areas, it is important to
identify the origin of this pollution in order to reduce and control its impact. Identifying the
sources of urban pollution and then assigning responsibilities enables a process to implement
measures and control air pollution. Assessing the responsibility or share of cities for their
pollution has important implications. For being effective, pollution reduction plans must be
designed and applied to target the most polluting sectors at the relevant spatial (national, regional
and/or local) and with the appropriate temporal scales. In this context, quantifying the share or
the city pollutions caused by their own emissions becomes a crucial element to determine
whether actions need to be applied locally or at the regional, national country or continental
scales. This has important governance consequences for the effective control of air pollution.
For pollutants like NO$_2$, that mostly originate from traffic sources and have a relatively short
lifetime in the atmosphere, there is a general agreement on the fact that cities are the main
contributor to this pollutant concentration levels and that acting locally on traffic emissions is the
most efficient way of improving NO$_2$ concentration levels in a particular city (Tobias et al.,
2020). There is available European-wide information such as in Degraeuwe et al. (2019)
providing overviews of the potential impact of traffic emission reductions per vehicle type in
different European cities. There is also agreement regarding O$_3$ that this secondary pollutant is
most effectively reduced by implementing reduction measures at larger spatial scales, involving
actions driven at the regional and even continental scales (e.g. Luo et al. 2020). For other
pollutants, like PM$_{2.5}$, complex physical and chemical atmospheric processes with different time
scales drive its formation, involving numerous precursors themselves emitted by several sources.
The sources of PM$_{2.5}$ pollution range from local traffic, domestic fuel burning and industrial
activities to regional sources such as agriculture in rural areas. Even though the latter emissions
do not originate from cities, Thunis et al. (2018) showed that their impact on urban pollution
could be important, reaching up to 30% in several European cities. Because of this complexity,
there is less consensus regarding the responsibility or share of a city to its pollution when





addressing PM$_{2.5}$. Because of this lack of consensus and the major burden of PM$_{2.5}$ on health, we
focus our analysis on this pollutant.
The usual approach to assess the city share to its pollution levels (in other words the city
responsibility) is source apportionment (SA). However, many SA approaches exist and many
ways to parameterize them as well, leading to a variety of results and interpretations. The most
widely used SA methods are the "potential impact" (or brute force), the "increment" and
"tagging" aproaches.  An overview description of these methods and an evaluation of their
limitations and capabilities for use can be found in Thunis et al. (2019). For the 18 million
inhabitant's city of New Delhi, Amann et al. (2017) concluded that only 40% of the PM$_{2.5}$
pollution was originating from local city sources, based on potential impacts SA and expressed
in terms of city averaged population exposure, averaged yearly. In the context of the Copernicus
programme, CAMS (Copernicus Atmosphere Monitoring Service) performs SA calculations
daily with two different approaches, namely tagging and potential impacts, for a series of
European cities. Results show important differences on a day-by-day basis although these
differences smooth out when considering longer term averages (Pommier et al. 2020). Based on
the increment approach, Kiesewetter and Amann (2014) derived SA estimates for a series of
European cities and aggregated these detailed results at country levels, leading to relatively low
city responsibilities (e.g. about 25% for French, German or Italian cities). Based on a potential
impact approach, Thunis et al. (2018) estimated city shares for 150 cities in Europe. They
highlighted their large variability across Europe and stressed the importance of the definition of
the city on the results, by testing the sensitivity to different city extensions. The choice of the SA
method but also the way this method is configured, can lead to very different outcomes for the
city share to its pollution, ranging from cities being a major contributor to their pollution to cities
having a limited responsibility. This explains why the actual city responsibility on its pollution is
yet discussed, and why some authors stress the importance of local actions (Thunis et al., 2018,
Wu et al. 2011, Raifman et al., 2020) when others stress the need for regional, national or even
continental actions (ApSimon et al. 2021, Liu et al., 2013). This diversity of conclusions has
serious consequences in terms of policy decisions. Blaming external (i.e. outside the city)
pollution sources as main responsible for urban pollution is sometimes an easy argumentation for
decision-makers to justify local inaction.
This work aims at explaining the main causes of discrepancies between different assessments of
the city emission's impact on its pollution levels and show that these discrepancies generally lead
to underestimating the city's responsibility. It proposes a specific harmonized nomenclature for
source allocation approaches, and it shows how it is important to document the choices to enable
correct interpretation of the results. We begin with a conceptual overview of the parameters
structuring any SA approach (Section 2). This includes the definition of the key parameters to
any SA study: indicator, source, receptor, and methodology to relate them. Then (Section 3) we
assess the sensitivity of the urban SA results to the choices of these four parameters. In Section
4, we analyze implications in terms of air quality planning and suggested strategies. We finally
provide conclusions in Section 5.

## 2. Assessing the city responsibility on air pollution: Main concepts

In this section, we detail the steps required to quantify the responsibility of a city on its air
pollution, through source apportionment (SA). SA is a methodology that serves to estimate the



contribution of a given source at a specific receptor for a given indicator (for example the
concentration of a given pollutant like PM or $NO_2$). It involves the following steps (Figure 1):
(1) defining a relevant indicator, denoted as (I) to characterize air pollution
(2) defining the receptor (R) through its spatio-temporal characteristics, i.e. the area $(\bar{x}_r)$
and time period $(\bar{t}_r)$ over which the indicator is averaged
(3) defining the source (S) through its spatio-temporal characteristics, i.e. the city area
$(x_s)$ and time period for which the city responsibility is assessed $(t_s)$
(4) selecting the source apportionment (SA) methodology to capture the processes that
relate the source to the receptor.

Figure 1 summarizes these steps, as well as the nomenclature and symbols used in this work. We
use this new nomenclature to attach contextual information (i.e. metadata) to the source
apportionment. Further explanations of the symbols are given in the subsections below.

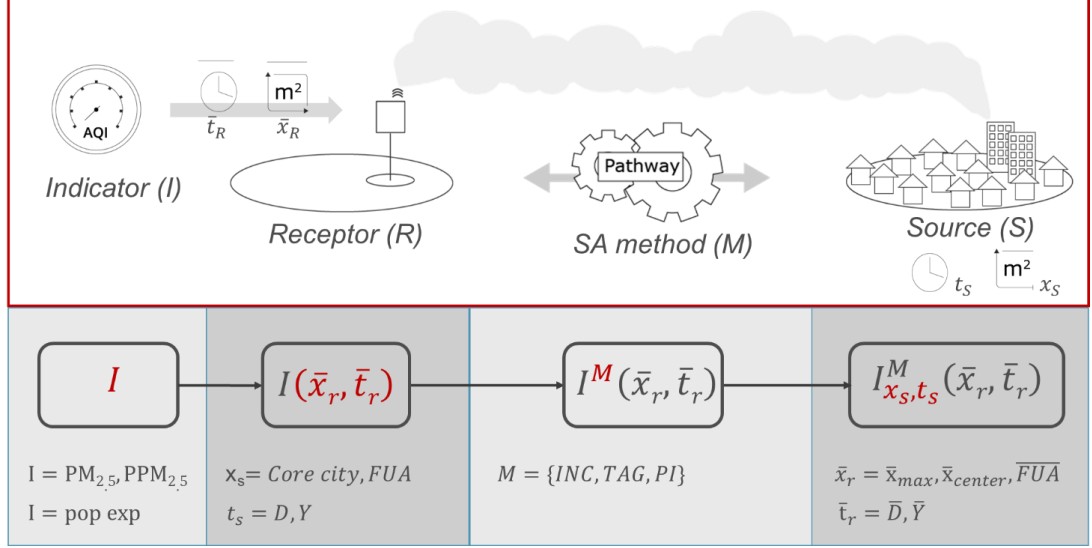

*Figure 1: Schematic flow chart representing the four steps required to fully define any SA process. The red letters indicate the*
*indicator characteristic under consideration. The general notation for the indicator (I) includes a superscript for the*
*methodological approach (M), a subscript to inform on the source (S) and brackets to inform on the receptor (R). The spatial and*
*temporal dimensions associated to the source and receptor are denoted by "x" and "t", respectively. The overbar indicates an*
*averaging process. The lowest row provides for each parameter examples used in this work.*

## 2.1   Definition of the air pollution indicator (I)

The first step required to assess the role/responsibility of city emissions with respect to its air
pollution, is to define an indicator that identifies the pollution aspect we are interested in. The
indicator can be defined in many ways. For example, as the total concentration of a given
compound (e.g. PM), or as a specific constituent of that total concentration (e.g. $PM_{2.5}$ or its
primary fraction, PPM), or as a composite based on a mix of different pollutants (e.g. maximum
among $O_3$, $PM_{2.5}$ and $NO_2$ concentrations as in some air quality indexes such as ATMO2003) or
as population exposure (i.e. product of population and concentration).

## 2.2 Definition of the receptor (R)

Estimating the indicator, either from a measuring instrument or from a model simulation, implies an averaging process, both in space and time. For model data, averages correspond to the spatial and temporal resolutions (e.g. the time step and grid cell size) whereas for measurement, the space-time average will depend on the instrument acquisition time and on the atmospheric dispersion characteristics at the measuring site. Regardless of these intrinsic time and space averages, indicators are generally averaged over longer spatial and temporal scales for convenience. The receptor is defined as the spatio-temporal entity over which the indicator is averaged. Both a spatial and a temporal scale (denoted by $\bar{x}_r$ and $\bar{t}_r$, respectively) must be associated to the receptor to define it.

For the temporal dimension, typical examples for $PM_{2.5}$ are days ($\bar{t}_r = \bar{D}$) or years ($\bar{t}_r = \bar{Y}$). Spatially, the indicator can be estimated at a specific location, e.g. the city center ($\bar{x}_r = \bar{x}_{center}$), at the location where the maximum concentration occurs ($\bar{x}_r = \bar{x}_{max}$) or averaged over the city ($\bar{x}_r = \overline{city}$). For convenience, we use indifferently the following notations to refer to the receptor:

$$R(\bar{x}_r, \bar{t}_r) = R = \bar{x}_r, \bar{t}_r \tag{1}$$

## 2.3 Definition of the source (S)

The source is defined as the spatio-temporal entity for which we assess the contribution to the indicator. For the purpose of this work, the source is defined as the city, and more precisely as the emissions that originate from a given city. The source emissions (denoted by E) are indeed responsible for the pollution fraction that can be associated to the source/city at the receptor (R). These emissions are characterized by a spatial ($x_s$ = extension of the city) and a temporal scale ($t_s$ = period of time over which the source activity is assessed). For convenience, we use indifferently the following notations to refer to the source:

$$S(x_s, t_s) = S = E = city = x_s, t_s \tag{2}$$

In this work, we analyse in particular the impact of the city extension ($x_s$) on the apportionment outcome. For this purpose, we define cities in two ways:

(1) as core cities, i.e. the local administrative units, with a population density above 1500/km$^2$ and a population above 50,000, where the majority of the population lives in an urban center and

(2) as functional urban areas (OECD, 2012, denoted as "FUA") composed as core cities plus their wider commuting zone, consisting of the surrounding travel-to-work areas where at least 15% of the employed residents work in the city.

Details on the FUA and core city areas are available for 150 EU cities in the urban $PM_{2.5}$ atlas (Thunis et al. 2017). Note that other city definitions exist. In the context of the CAMS source allocation analysis, city are defined as an arbitrary number of grid cells in the modelling domain (Pommier et al., 2020).



Finally, we define the city background as the sum of all contributions from sources that are not
covered by the spatial ($x_s$) and temporal ($t_s$) scales of the city source.
One main difference between sources and receptors is that for the latter, spatio-temporal
characteristics are averaged. Apart from this, temporal and spatial characteristics can also differ
in terms of value. For example, the source can be defined as the FUA ($x_s$ = FUA) while the
receptor is a specific location ($\bar{x}_r = \bar{x}_{max}$). Temporally, interest can be on assessing the
contribution of the city weekly activity ($t_s$ = 1 week) for a given day ($\bar{t}_r = \bar{D}$) at the receptor. In
the results presented here, the source and receptor temporal scales are however chosen identical
for convenience.

## 210    2.4    Selection of the SA methodology

When the air pollution indicator and the spatio-temporal characteristics of both the receptor and
the source have been selected, the next step consists in distinguishing and quantifying the
fractions of the indicator related to the city source ($I_{city}(R)$) and to the background ($I_{bg}(R)$) at
receptor R, respectively. This decomposition is summarized by the following equation:

$$I(R) \rightarrow \{I_{city}(R), I_{bg}(R)\} \tag{3}$$

Different SA methodologies exist to perform this operation. In this section, we describe three
main approaches but only in brief, as details about each of these are discussed in other works
(Clappier et al. 2017; Thunis et al., 2019, 2018; Mertens et al. 2018). As mentioned previously,
we use the indicator's superscript to refer to its calculation method [$I_{city}^{M}(R)$]. Methods are
summarized in Table 1.
Potential impacts (PI): The city contribution in this method is denoted as $I_{city}^{PI100}(R)$ and is
calculated as the difference between two simulations: a base-case that includes the city
[$I(R)$] and a scenario in which the city emissions are switched off [$I_{city^{100}}(R)$]. In this notation,
the source superscript (here, 100) indicates the percentage intensity by which the source
emissions are reduced. Reductions are intended as percentage variations from the base-case
situation.  The same approach can be used with reduction percentages that are lower than 100%.
In this case the resulting difference is divided by the reduction percentage to obtain the potential
impact ($I_{city}^{PI\alpha}(R)$). A similar approach is used to calculate the background contribution, i.e. by
removing or reducing partially the background emission sources. Potential impacts methods for
source apportionment are widely used (Osada et al. 2009; Huang et al. 2018; Wang et al. 2014;
Wang et al. 2015; Van Dingenen et al. 2018; Thunis et al. 2016; Clappier et al. 2015; Pisoni et al.
2017).
Increment (INC): With this methodology, the background contribution is estimated as the
concentration observed/modelled at a given location "y" [$I_{bg}^{INC}(R) = I(\bar{y}, \bar{t}_r)$]. This location must
be far enough from the source, not to feel its influence but be close enough to the source to avoid
influences from other sources, external to the city. These assumptions are further described and
discussed in Thunis et al. (2017). The city contribution is then obtained as the difference between
the base case indicator and the background contribution [$I_{city}^{INC}(R) = I(\bar{x}_r, \bar{t}_r) - I(\bar{y}, \bar{t}_r)$]. The
increment methodology has been used e.g. by Lenschow et al. (2001), Petetin et al. (2014),





Kiesewetter et al. (2015), Squizzato et al. 2015, Timmermans et al. 2013, Keuken et al. 2013,
Ortiz and Friedrich 2013 and Pey et al. 2010.

Tagging (TAG): With this approach, species emitted by the city are numerically tagged and
followed through the modelled transport, dispersion and chemical transformation processes.
When chemical transformations take place, preserved atoms are used as tracers. For example, the
nitrogen atom (N) will be used to follow the NO source emissions through its successive
transformations into $NO_2$ and $HNO_3$ to reach its final product $NO_3$, that will then be attributed to
that source. Example of tagging applications are e.g. Kranenburg et al. 2013, Yarwood et al.
2004; Wagstrom et al., 2008; Kwok et al. 2013; Bhave et al. 2007; Wang et al., 2009. Some of
these approaches are implemented operationally to estimate daily city contributions on air
pollution (https://topas.tno.nl/documentation/).

The formulations corresponding to these three main approaches are summarized in Table 1.

A few key points are worth noting. While tagging and potential impacts approaches explicitly
consider city emissions in their calculations, this is not the case for increments that only refer to
them implicitly. By construction, both the increment and tagging approaches are additive [i.e.
$I(R) = I_{city}(R) + I_{bg}(R)$] whereas this is not the case for potential impacts when pollutants
behave non-linearly because of air transport, deposition or chemical processes (Clappier et al.,
2017).



| | **City contribution** | **Background contribution** |
|---|---|---|
| **Potential Impact** | $I_{city}^{PI\alpha} = \dfrac{I(R) - I_{city^{\alpha}}(R)}{\alpha}$ | $I_{bg}^{PI\alpha} = \dfrac{I(R) - I_{bg^{\alpha}}(R)}{\alpha}$ |
| **Increment** | $I_{city}^{INC} = I(\bar{x}_r, \bar{t}_r) - I(\bar{y}, \bar{t}_r)$ | $I_{bg}^{INC} = I(\bar{y}, \bar{t}_r)$ |
| **Tagging** | $I_{city}^{TAG} = \displaystyle\sum_{E}^{city} I_E(R)$ | $I_{bg}^{TAG} = \displaystyle\sum_{E}^{bg} I_E(R)$ |

*Table 1: Formulation of the three main methods to estimate the contribution/impact/increment of a city. The letters, I, S and R*
*refer to the indicator, source and receptor, respectively. The indicator superscript refers to the SA method (PI for potential*
*impacts, INC for increments and TAG for tagging) while its subscript indicates the source (city or background (bg)). α represents*
*the percentage reduction factor applied for the source emissions in the potential impacts method. See text for additional details.*

## 270   3. Results

Recognizing the impossibility of assessing the sensitivity of the results for all combinations of
indicators, source, receptor and methodology, we focus our analysis on comparisons in which





only one parameter is changed at a time, to highlight major sensitivities. For this purpose, we use
the following two main sources of data and results.
• SHERPA: SHERPA is a modelling tool, based on Source-Receptor Relationships that
represent a simplified version of a Chemistry Transport Model, used to simulate the
contribution to $PM_{2.5}$ concentration levels by all precursor emissions ($NO_x$, NMVOC,
PPM, $SO_2$ and $NH_3$) from different cities in Europe (Clappier et al. 2015, Thunis et al.
2016, 2018). In its current configuration, SHERPA is based on the CHIMERE model
(Menut et al. 2013) covering the whole of Europe at roughly 7 km spatial resolution. In
this work, we use the source apportionment results over 150 cities as reported in the
PM2.5 urban atlas (Thunis et al., 2017) as well as additional SHERPA data to provide
further analysis.
• EMEP simulations: The EMEP model is an off-line regional transport chemistry model
(Simpson et al., 2012; https://github.com/metno/emep-ctm). The model has 20 vertical
levels, with the first level around 50 m. The model uses meteorological initial conditions
and lateral boundary conditions from the European Centre for Medium Range Weather
Forecasting (ECMWF-IFS). The meteorological year is 2015. Detailed information on
the meteorological driver, land cover, model physics and chemistry are described in
Simpson et al. (2012) and in the EMEP Status Report 2017
(https://emep.int/publ/reports/2017/EMEP_Status_Report_1_2017.pdf). In this work, we
use specific simulations where emissions have been removed partially or fully in a series
of European cities. Additional details regarding these simulations are provided together
with the discussion of the results.
Based on these sources of information and data, we discuss hereafter the sensitivity of the SA
results to the choice of the indicator (Section 3.1), to the choice of the methodology (Section
3.2), to the source (Section 3.3) and finally to the receptor (Section 3.4).

## 3.1 Sensitivity to the indicator

The implications resulting from the choice of the indicator are illustrated in Figure 2 for four
indicators, based on SHERPA results for 150 cities in Europe. The four indicators selected to
characterize air pollution are: a) the $PM_{2.5}$ concentration (top left, from Thunis et al. 2017), b) the
anthropogenic fraction of $PM_{2.5}$ ("$PM_{2.5}$ ant", top right), c) the primary anthropogenic fraction of
$PM_{2.5}$ ("$PPM_{2.5}$ ant" bottom left) and d) the primary fraction of $PM_{2.5}$ originating from the
transport and residential sectors ("$PPM_{2.5}$ oxy", bottom left). The reference ($PM_{2.5}$ total mass, top
left) corresponds to the indicator currently used in legislation (e.g. European Ambient Air
Quality Directive, AAQD2008) against which health impacts are correlated (WHO2005). In the
second case, the indicator is limited to its anthropogenic fraction ($PM_{25}$ ant), excluding therefore
natural contributions (dust, marine salt…). This is motivated by the fact that policies have no
impact on this component. According to this indicator, city contributions increase significantly
(by about 20% in average) and in some cities where natural dust pollution is important (e.g. in
Sicily), the city responsibility shifts from minor to major. If we further restrict the indicator to its
primary anthropogenic fraction ("$PPM_{2.5}$ ant", bottom right) because of its suggested higher
health burden (Park et al., 2018; Viana et al., 2008), the city contribution then increases



significantly in most cities. This becomes even more striking if we limit the indicator to the
PPM$_{2.5}$ fraction originating from the transport and residential sectors (bottom right). These two
sectors have recently been shown to generate the largest burden on human health given the high-
oxidative potential of their emissions (Rankjar et al., 2020, Li et al. 2016). With this indicator,
the majority of EU cities become main contributors to their pollution. Regarding the latter
indicator, it is important to note that although the increasing adoption of electric vehicles shows
rather positive impacts on health (Choma, 2020), the remaining PM emissions from road traffic
like tires and brake and road wear emissions (Kole et al., 2017; EC, 2014; Ntziachristos and
Boulter, 2019) will remain an issue. The calculation of various geochemical indices (enrichment
factor, geo-accumulation index, pollution index and potential ecological risk) also show that road
dust is extremely enriched and contaminated by elements from tire and brake wear (e.g. Sb, Sn,
Cu, Bi and Zn).

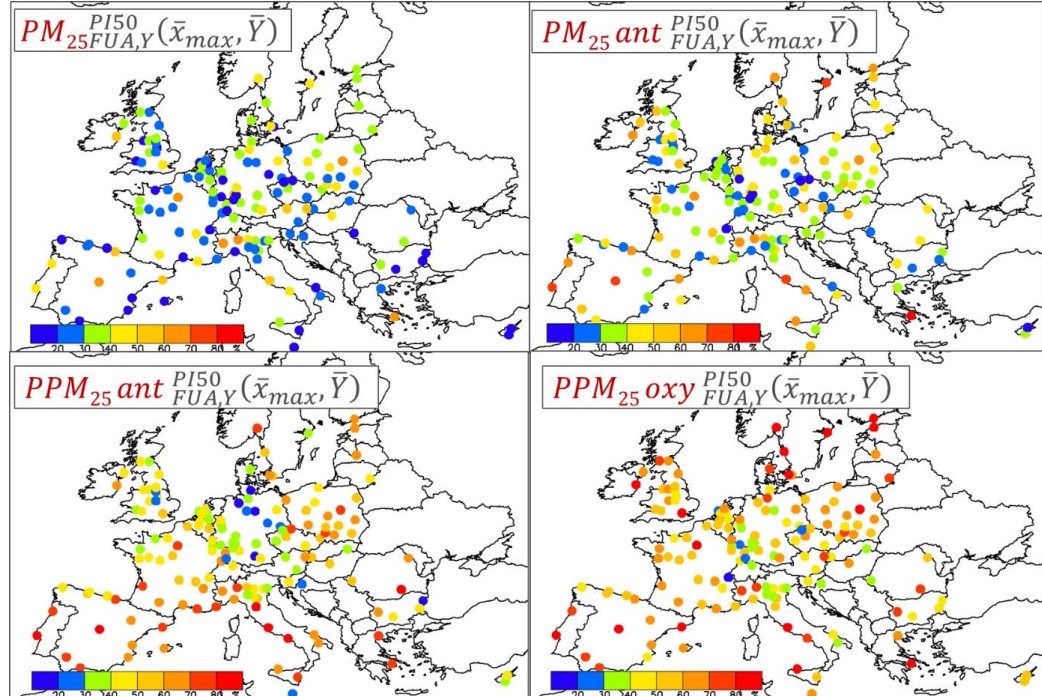

*Figure 2: SHERPA results for 150 major cities in Europe for the overall PM2.5 concentration (top left), for its anthropogenic*
*fraction ("PM25_ant", top right), for its anthropogenic primary fraction ("PPM25_ant", bottom right) and for its primary*
*fraction originating from the transport and residential sectors ("PPM25_oxy", bottom left). For all cities, the source is defined*
*spatially as the FUA over which emissions are reduced over a year (Y). The receptor is defined as the city location where the*
*concentration is maximum ($\bar{x}_{max}$) and the indicator is averaged yearly at the receptor ($\bar{Y}$). All calculations are made with the*
*same SA methodology, namely, potential impacts (PI) with city emissions reduced by 50% (PI50)*

## 3.2   Sensitivity to the SA methodology

A comparison of SA methodologies is proposed in Thunis et al. (2019) where the potential
impact, increment and tagging approaches are compared both on simple theoretical examples and
on real data to highlight differences among methods and stress their limitations. In this section,
we summarize the main findings of this work and complement it with comparisons that focus on



the apportionment of the city vs. background contributions. We also provide in the appendix a
comparison of all SA methods discussed in this section, applied on a theoretical example tuned
to the city scale.

Increment vs. potential impacts

Thunis (2017) compared increments and potential impacts with the SHERPA model for a series
of European cities. They showed that increment approaches lead to important underestimations
(30 to 50%) of the city responsibility for $PM_{2.5}$ and $NO_2$ with respect to potential impacts. This
underestimation is explained by the non-fulfilment of the two underlying increment assumptions,
related to the external location [i.e. y in $I_{bg}^{INC}(R) = I(\bar{y}, \bar{t}_r)$] that must: 1) be far enough from the
city, not to feel its influence but 2) close enough to the city to avoid influences from sources
external to the city. The Authors show that these two assumptions are seldom fulfilled in reality.

Tagging vs. potential impacts

Clappier et al. (2017) discussed the concepts underlying these two SA methods and showed that
important differences in terms of results arise as soon as non-linear processes are present. Belis
et al. (2020) highlighted and quantified these large differences based on a real-case inter-
comparison exercise. Finally, Thunis et al. (2019) reviewed in their work many inter-
comparisons between tagging and potential impact SA results. In their application over the Po
basin (Italy), they showed that differences are large for the agriculture sector (dominated by $NH_3$
emissions) but are also important for other sectors, when dealing with high temporal resolution
(e.g. daily) at the receptor. Unfortunately, these examples did not address the particular case of a
city scale apportionment.

Full vs. partial potential impacts

To analyze differences between full and partial impacts, we use a series of EMEP simulations in
which we remove totally (PI100) or partly (PI20) the London FUA emissions (source) during an
entire year. Figure 3 shows the differences between city contributions obtained with the two PI
methods. Differences can be important (up to 25 percentage points for specific days). Although
the number of high-difference days is limited (leading to a yearly average difference of few
percents), these days might represent high pollution episodes for which assessing the city
responsibility is important to act. In general, the higher resolution applied to the temporal and/or
spatial averages at the receptor, the largest the differences are among methods. It is also
interesting to note that partial potential impacts systematically underestimate full potentials (no
negative values).

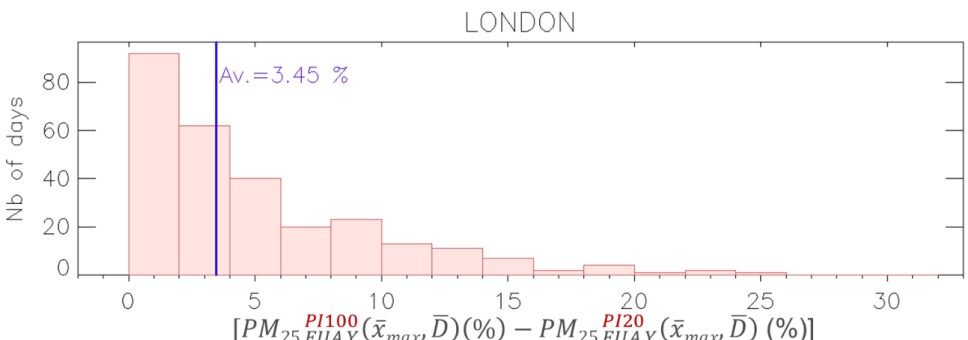

*Figure 3: Histogram of daily city contribution differences to London PM$_{2.5}$ levels between two potential impacts methods, PI100 and PI20, calculated with the EMEP model. The source is defined spatially as the FUA where emissions are reduced yearly (Y subscript). The receptor is defined as the city location where the maximum yearly averaged concentration is modelled ($\bar{x}_{max}$), and temporally as daily average ($\bar{D}$). Each column represents the number of days with a specific PI difference (PI100 - PI20). The blue line provides the yearly average difference.*

### 3.3  Sensitivity to the source

Figure 4 shows the comparison between SA obtained with sources defined as core cities (left) and as FUA (right). The city contribution / responsibility is multiplied by a factor 2 on average (see also Figure 8) when FUA are considered. The larger spatial extension of the FUA and its implied additional emissions explain the differences that lead some cities to become a major actor, i.e. where the city contribution dominates the background one (e.g. Athens, Warsaw, Milan, Turin and Rome).

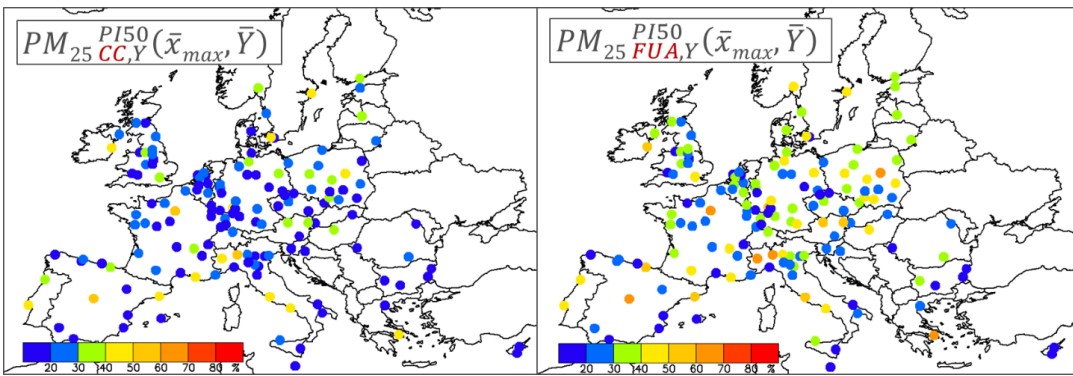

*Figure 4: Maps of city contributions obtained for spatial sources defined in 2 ways: core city (CC, left) and FUA (right). Results are shown for 150 cities in Europe, based on the SHERPA-CHIMERE model using a potential impact SA method for a reduction strength of 50% (PI50). The indicator is the total PM$_{2.5}$ concentration. The receptor is selected as the location where the maximum yearly average concentration occurs ($\bar{x}_{max}$) and applies yearly time average ($\bar{Y}$). The source emissions are reduced over a full year (Y).*

### 3.4  Sensitivity to the receptor

In this section, we discuss the spatial and temporal averages applied at the receptor. Spatially, different averaging options exist, ranging from a single location (i.e. one modelling grid cell) to more or less extended areas covering part of the source or even larger. To illustrate the


sensitivity of SA to that choice, we use the case of Paris (Figure 5) where emission have been
reduced over the FUA (source) over a full year.
SA varies largely from one location to another within Paris. We highlight this with bars that
distinguish the city vs. background contributions for locations at different distance from the city
centre. We note opposite trends, dominated by the city source (around 60%) at the city center
and dominated by the background source towards the periphery (around 80%). While the SA at
the city centre is representative of a single cell within the city, this is not the case for SA close to
the periphery. This is highlighted by the city rings (below the X-axis) that indicate the area of
representativeness of a given SA. When we average spatially an indicator ($PM_{2.5}$ or population
exposure) over a receptor that covers the entire FUA (all 6 rings), these areas of
representativeness enter into play. The brown curve indicates the weight (in the spatial average)
attached to each city ring, relatively to the city total (i.e. all rings). Weights increase fast when
moving towards the periphery because of the larger ring areas. The spatial averaging process
leads to over-representing the periphery, which overweight the city center SA by almost a factor
40. It is interesting and counter-intuitive to note that with this averaging process, the city
responsibility decreases when the city area increases. With population exposure as indicator
(weights shown by red curve), the rapid population density decrease balances the ring area
increase when moving outward, leading to weights that dominate for middle rings. It is
interesting to note, that with average population exposure, the city center weight is yet similar to
the weight obtained 28 km away.

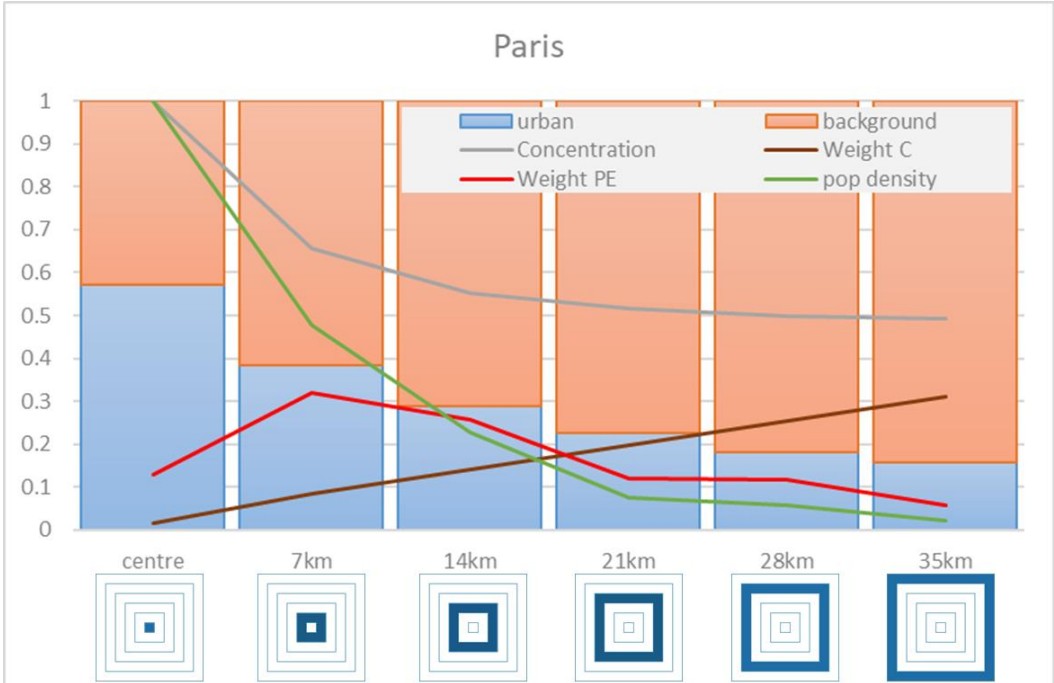

*Figure 5: City rings' source apportionment for Paris $PM_{2.5}$ and associated population exposure. The city/background*
*apportionment (bars) is represented for rings (i) progressively more distant from the city centre (X axis). The ring average*
*concentration ($C_i$) and population density ($P_i$) relative to the city centre values are represented in blue and green, respectively.*





*The relative (to the FUA total, i.e. all rings) weight of each ring (i) in the city average concentration (brown) is calculated as*
$C_i * S_i / \sum_i (C_i * S_i)$ *where $S_i$ is the ring area, respectively. A similar expression: $C_i * S_i * P_i / \sum_i (C_i * S_i * P_i)$ is used to determine*
*the weight of each ring in the calculation of the average population exposure (red curve).*
Figure 6 compares SA for 150 cities obtained for receptors defined (1) as the location where the
maximum concentration is reached within the FUA ($\bar{x}_{max}$) and (2) as the FUA spatial average
($\overline{FUA}$). In average, city impacts for a spatially averaged receptor are about 55% lower.
Depending on the spatial characteristic of the receptor, some cities will be considered as minor or
major actors with respect to their pollution. We discuss this issue further in Section 4.

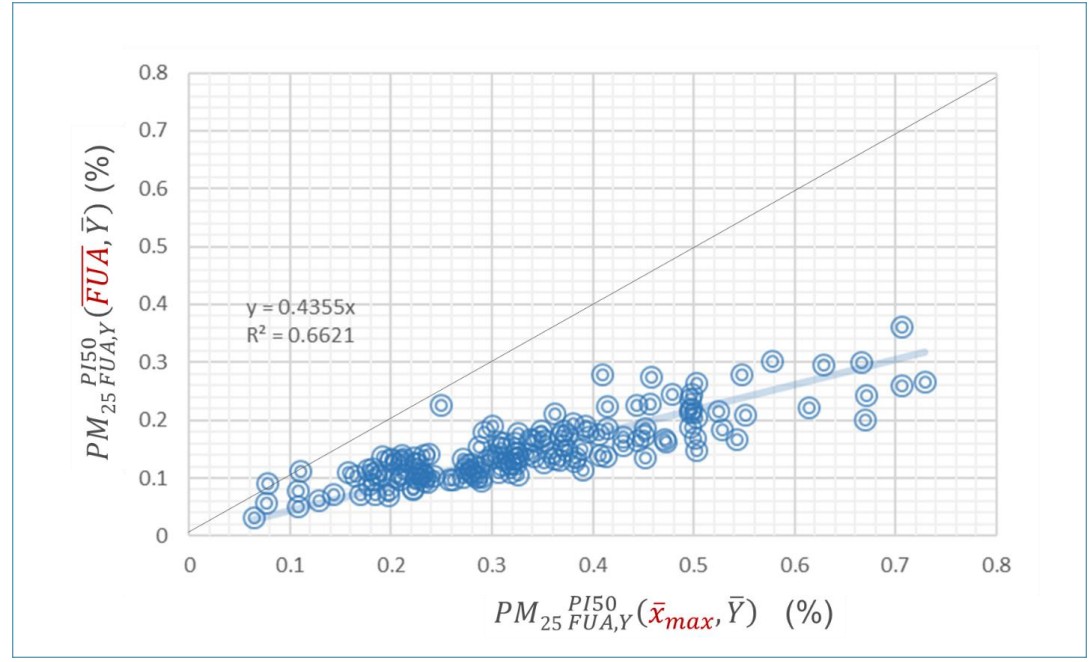

*Figure 6: Comparison of potential impacts for 150 cities in Europe obtained for a receptor spatially defined as the location where*
*the concentration is maximum in the city ($\bar{x}_{max} - X$ axis) and defined as the FUA spatial averaged ($\overline{FUA}$). For these calculations,*
*the source are defined as the FUA over which emissions are switched off during the whole year. The indicator is the total $PM_{2.5}$*
*mass. All results are based on the SHERPA-CHIMERE model using a potential impact SA method for a reduction strength of 50%*
*(PI50) and are based on yearly averages at the receptor ($\bar{Y}$).*
As seen from these results, spatial averages at the receptor significantly reduce the city
responsibility, potentially leading to underestimating the city ability to reduce pollution levels
via local controls. The large differences resulting from the choice of the receptor settings prevent
meaningful comparisons. It is for example challenging to compare CAMS city contributions that
are averaged spatially over the city area with the urban results obtained in the context of the
Thematic Strategy on Air Pollution (Kiesewetter and Amann 2014) that are aggregated at
country level or with SHERPA estimates based on a single grid cell receptor. It is therefore
crucial to associate all SA settings (metadata) to the results in order to inform on the
meaningfulness of a comparison. We discuss further this issue in the context of air quality
planning in Section 4.



Similar considerations apply to temporal averages. Figure 7 compares SA obtained when the
indicator at the receptor is averaged yearly and seasonally with daily single values. For a yearly
average, Madrid city's contribution is 54% but the spectra of daily contributions show variations
that range from 10 to beyond 90%. Even seasonal averages show important differences with a
factor 2 between summer and winter. Similarly, to spatial averages, temporal averages
encompass a large spectra of SA outcome. Indicators averaged yearly at the receptor have been
used for example in SHERPA (Thunis et al. 2017), GAINS (Kiesewetter and Amann, 2014)
whereas daily indicators are used in CAMS (Pommier et al., 2020).
Note that spatial averages have a larger smoothing effect than temporal ones because they are
bidimensional.

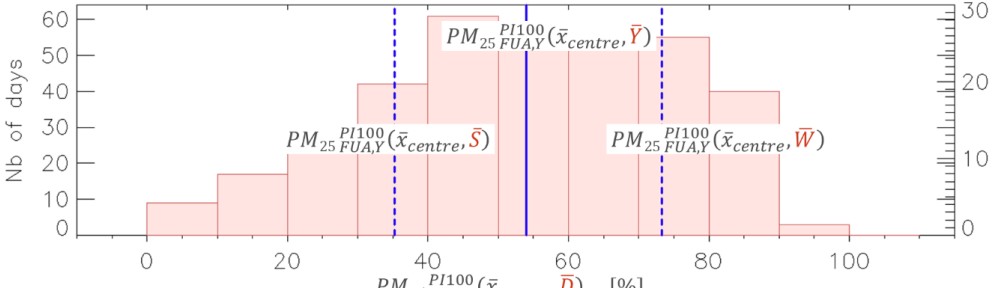

*Figure 7: Frequency histogram of daily potential impact at 100% (PI100) modelled with the EMEP model for the city of Madrid.*
*Each column represents the number of days with a given daily PI. The blue line provides the yearly average PI. For these*
*calculations, the source is the Madrid Functional Urban Area (FUA) over which emissions are switched off during the whole year*
*(Y). The indicator is the total PM$_{2.5}$ mass. The receptor point is the city centre location ($\bar{x}_{centre}$).*

## 3.5 Assumptions and uncertainties

Most SA methods rely on models and are therefore characterized by a set of common strengths
and weaknesses. One of the main limitations attached to models is the spatial resolution and its
potential impact on the calculation of the city contribution. While a coarse resolution might be
able to capture relatively well the background (characterized by smoother fields), this will not be
the case for peak concentrations within the city. The coarser the model spatial resolution, the
largest the underestimation of the city responsibility will be (De Meij et al., 2007).
Uncertainties may also result from our incomplete knowledge of some model input parameters,
in particular chemical processes and emission sources. Some urban emission sources are not well
documented and are probably underestimated. This is the case of residential emissions for which
the inclusion of condensable remains a question mark (Bessagnet and Allemand, 2020, Simpson
et al., 2020) or for the resuspension of particles generated by vehicles (Amato et al., 2014).
These lacking or incomplete emission sources will lead to a potential underestimation of the city
responsibility as well.
In the next section, we discuss the consequences of these results on policy, in particular when SA
information is used to design air quality plans.





## 4. Implications for air quality strategies

Estimating the contribution of a city to its pollution has important consequences in terms of air quality management. Indeed, an important city contribution will be a logic argument to support substantial control measures at the local level to abate pollution. The effectiveness of the control measures then relies on the relevance and accuracy of this city contribution; over- or under-estimated city contributions potentially leading to inefficient measures.

In previous sections, we have seen that the city contribution largely varies depending on the choices made for the SA setting parameters (definition of the indicator, source, receptor and methodology), hence the challenge to obtain a relevant and accurate estimate to support local action.

Given the range of possible SA options and their impact on results, the first recommendation is obviously to report these SA setting choices together with the results to provide policymakers with the full picture and allow them to take informed decisions. This advocates for the use of the proposed nomenclature or a similar one that documents for the choices in the SA approach, providing accountability to the method and enabling correct interpretation of the results. The proposed nomenclature can be understood as a documentation of the SA metadata information. Apart from this point on the importance of documenting SA approach choices, we show below that some of the SA settings are fixed by the purpose of the study. We provide suggestions for the remaining free choices.

The recommended SA method is potential impacts (PI)

It is important to recall that not all SA methodologies are equally suited to support air quality planning. As mentioned by several authors (Burr and Zhang 2011, Qiao et al. 2018, Mertens et al. 2019, Clappier et al. 2017, Grewe et al. 2010, 2012; Thunis et al. 2019), potential impacts are recommended when non-linear species are involved (which is the case for $PM_{2.5}$ and $PM_{10}$ but also for other species like $NO_2$ or $O_3$). It is worth reminding that tagging or incremental approaches are yet erroneously used and believed to be suited for air quality planning purposes (Qiao et al. 2018; Guo et al. 2017; Itahashi et al. 2017; Timmermans et al. 2017; Wang et al. 2015, Hendriks et al. 2013). Although challenging practical issues are attached to potential impacts and may be seen as a burden (e.g. lack of additivity, see Appendix), they only reflect the complexity of the real processes that must be accounted for. Although uncertainties associated to the PI approach (e.g. imperfect emission inventory), may lead other SA methods to perform better in some instances because methodological biases compensate uncertainties, this is however coincidental. While uncertainties can be tackled and reduced to improve the approach, this is not the case of methodological biases. These points were extensively discussed in Thunis et al. (2019).

For the remaining of this section focusing on policy aspects, only potential impact results are discussed. Fixing the methodology however still leaves free options in terms of indicator, receptor and source. This is visualized in Figure 8 that summarizes the variability of the SA results presented in the previous sections (i.e. Figure 2,Figure 4 and Figure 6) for the 150 cities to these possible choices. Differences in terms of city responsibility reach a factor 2 in average for each of these remaining parameters with much larger values for some cities.



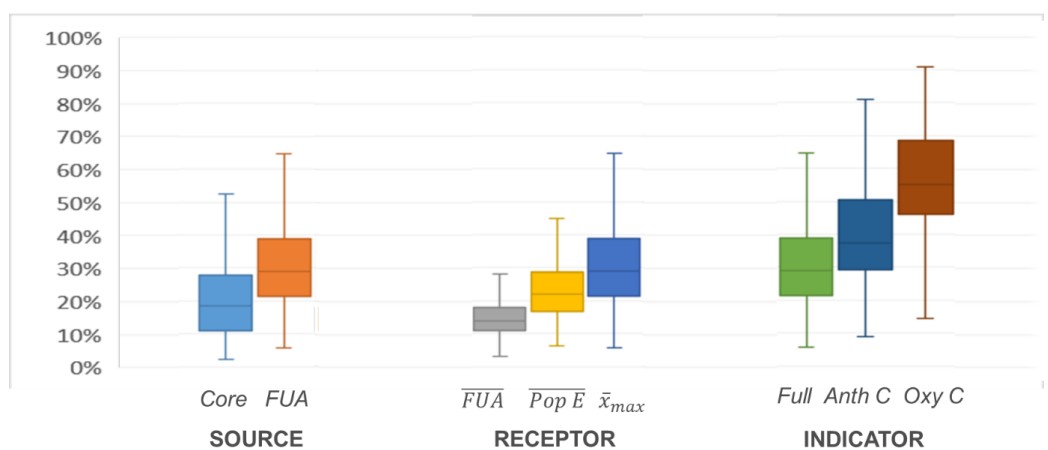

*Figure 8: Box quantile diagrams summarizing the city contributions to PM$_{2.5}$ levels for the 150 EU cities. All results are based on*
*a similar method (potential impacts at 50%), a similar temporal receptor ($\overline{Y}$) but for different choices of city sources (left),*
*receptors (centre) and indicators (right). See previous sections for details. The two extremities of each vertical line represent the*
*10$^{th}$ and 90$^{th}$ percentile contributions among the 150 cities, respectively. The box crossing horizontal line represents the median.*
INDICATOR: The indicator choice is driven by health and environmental objectives
The choice of the indicator is generally motivated by health or environmental considerations.
Currently, the WHO guidelines (WHO2005) refer to the total PM$_{2.5}$ mass as the indicator
correlating best with health impacts. These guidelines (or the AAQD limit values) are then the
logical and most relevant indicator choice among the options presented in Section 3.1 and shown
in Figure 2. As illustrated by Figure 8, evolving knowledge on health-related pollution impacts
(i.e. the increased toxicity of some PM$_{2.5}$ constituents like those related to the traffic and
residential activities) might however, drive the choice towards more detailed indicators (e.g.
PPM$_{2.5}$) leading to an increased responsibility for the cities.
SOURCE: Importance of matching sources with governance levels
Figure 8 shows that plans limited to city cores would be significantly less efficient than if applied
at the FUA scale. In average over all cities, the efficiency decreases by a factor 2 but larger
differences occur in many cities. The source does however not represent a free choice in the
context of policy practice. Indeed, authorities in charge of AQ plans only have power to act on
the area under their responsibility, which sets where measures apply. The same applies for the
source temporal characteristic, fixed as the period of time during which measures apply. A good
match between the SA settings and the temporal and spatial characteristics of the source is
therefore important to provide meaningful support to policy makers.
RECEPTOR: Drawbacks associated to spatial and temporal averaging processes at the receptor
As clearly shown in Figure 5, spatial averaging processes lead to a loss of information. In our
example, a city average based SA would totally occult the city center SA. It would lead to a
strategy that mostly targets the background at the expense of the city center, where the high





concentration issues would not be solved. This is well illustrated by Amann et al. (2017) who
analyse the responsibility of the city of New Delhi on its air pollution, both at a city center hot-
spot receptor and in terms of city average population exposure. In the first case, SA suggests
acting on local sources while in the second SA suggests acting on regional sources. Spatial
averaging drives the balance towards regional actions that will less effective in solving the
pollution issue at the city center.  The larger the city, the more important this shift will be. As
illustrated by Figure 8, there is more than a factor 2 between city-averaged and hot spot
indicators. Similar considerations apply to temporal averages.  Figure 7 clearly shows that yearly
average values hide the potential for effective local actions during wintertime and even more on
specific days.
Averaging implies merging, into one single number, locations and time instants that are
characterized by different and sometimes opposite SA. This may lead to strategies that will not
be efficient everywhere all the time. Whenever the final objective is to reduce a temporally
or/and spatially averaged indicator (e.g. average population exposure), strategies would gain in
efficiency with the following process: (1) perform SA and hierarchize the raw (not averaged) SA
results into homogeneous spatio-temporal clusters; (2) design strategies on the basis of these
clusters; (3) assess the strategy efficiency against the averaged indicator. The key is here to
design strategies on raw or clustered results rather than on averaged ones, to prevent information
loss.
Note that designing a unique strategy based on multiple SA results (point 2 above) does not
necessarily complicate the analysis, as these different SA will likely suggest action on different
sectors of activity that can be combined at the final strategy.

## 5. Conclusions

Although air quality has improved in Europe over the last decades, in great part thanks to
effective measures and consistent EU-wide legislation, pollution hot spots yet remain in many
European cities. The extent by which city emissions are causing these elevated urban pollution
levels is however still a subject of scientific discussion. Source apportionment represents a useful
technique to quantify the city responsibility but the approaches and applications are however not
harmonized, therefore not comparable, resulting in confusing and sometimes contradicting
interpretations.
In this work, we analyzed how different SA approaches apply to the urban scale and how their
building elements and parameters are defined and set. We identified the possible settings
associated to four key steps in SA: indicator, receptor, source and methodology. We showed that
different choices for these settings lead to very large differences in terms of results. In average
over the 150 European large cities selected as example, the choices made for the indicator, the
receptor, and the source each lead to an average factor 2 difference in terms of city
responsibility. These various options and the large differences that result, highlight the difficulty
of comparing results from different studies and stress the need to document the SA approach
with its related metadata – that documents the choices made for the key four steps.





This work advocates for the use of a harmonized nomenclature to support the comparability of
SA approaches. We propose the use of indexes and subindexes attached to the 4 key steps in any
SA approach in a harmonized way to uniquely document the approach and enable correct
interpretation of the results. We believe that the adoption of this nomenclature will provide
clarity to the scientific discussion on different results and enable the correct interpretation of the
results for policy applications. Even though this is applied to the specific case of $PM_{2.5}$, the
concepts presented here can easily be generalized to other pollutants.
In the context of supporting urban air quality plans, the SA configuration and most setting
parameters are driven by the purpose of the AQ plan itself and by its associated constraints.
While environmental and/or health related considerations guide the choice of the indicator, the
spatio-temporal characteristics of the source are strongly correlated to governance aspects. In
other words, the source characteristics should reflect the governance levels to facilitate
interpretation. Finally, the recommended SA method should be based on "potential impacts", to
prevent misleading interpretations in terms of expected AQ plan outcome.
At the receptor level, temporal and spatial averaging processes lead to a loss of information,
especially when diverging SA results are aggregated into a single number. Averaging process, in
particular spatial, often lead to favor strategies that target background sources while neglecting
actions that would be efficient at the city center. In our 150 cities example, the impact of spatial
averaging leads to an average factor 2 difference in terms of city responsibility. Not only results
differ from one city to the other, and from one location to another in a given city, they also differ
through time. To cope with this variability, we recommend using non-averaged SA results for the
design of AQ strategies. Once clustered in homogeneous spatio-temporal classes, these can serve
to understand where and when actions are most efficient. When implemented, the efficiency of
abatement measures can then be assessed via spatially and temporally averaged indicator (e.g.
city average population exposure).
The responsibility of a city to its pollution is obviously city dependent. But even for a given city,
SA studies using different approaches and parameter settings will deliver very different
outcomes. It is important to note that a departure from the methodological recommendations
listed above, additional uncertainties and assumptions will most often lead to a systematic and
important underestimation of the city responsibility. We showed that in average over 150
European cities, departures in terms of source, receptor, and indicator may lead for each to a
factor 2 underestimation. This comes with important implications: if cities are seen as a minor
actor, plans will target in priority the background at the expense of potentially effective local
actions.
Future work will consist in comparing spatially/temporally averaged SA results with SA results
that are clustered in homogeneous spatio-temporal classes and assess the implications in terms of
AQ strategy.

## Acknowledgements

The Authors thank Chloé Thunis for her support with the infographics



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





## Appendix A

To illustrate the differences among SA methods, we use here the theoretical example
schematically represented in Figure . A city source (in red) emits with a Gaussian dispersion
profile both primary PM (PPM) and a gas-phase precursor ($NO_x$). The background pollution (in
blue) is composed of a mix of $NO_x$, $NH_3$ and PPM compounds. The various chemical reactions
that take place are simplified here for convenience into a single reaction. One mole of $NH_3$ reacts
with one mole of $NO_x$ to create one mole of ammonium nitrate ($NH_4^+NO_3^-$), i.e. secondary PM.
($NO_x + NH_3 + X \rightarrow\rightarrow NH_4^+NO_3^-$). We assume here that the external compounds involved in the
reaction (X) are abundant and do not have a limiting effect on the formation of PM. While the
city emissions (source) remain unchanged, we modify the relative importance of the three
background compounds so that the background becomes in turn PPM, $NO_x$ and $NH_3$ dominated.
The PM concentration at a given location "x" is given by:

$$PM(x) = PPM(x) + min\{NO_x(x), NH_3(x)\}_{mole} \times NH_4^+NO_3^- \qquad (4)$$

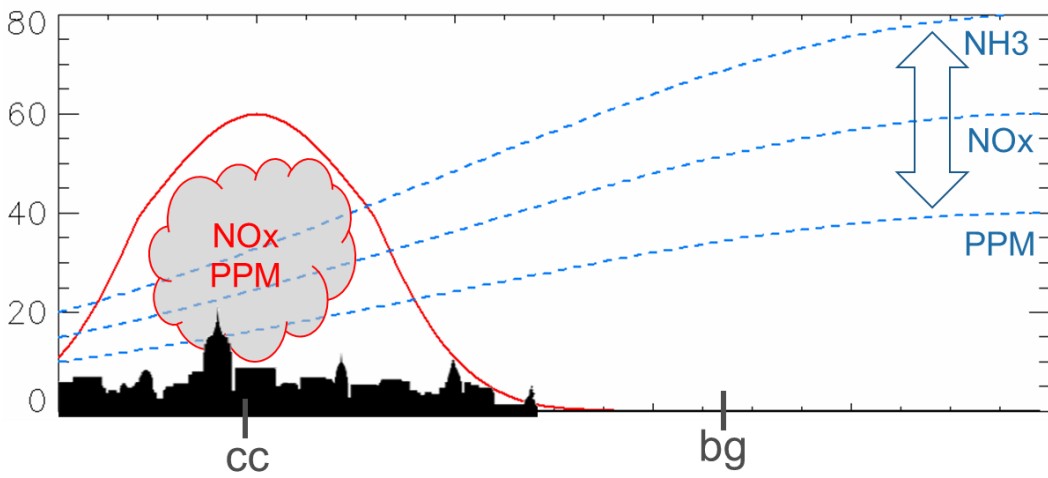

*Figure A1: Schematic representation of the theoretical example used to compare the three SA approaches. The city source (in red) emits $NO_x$ and PPM. The background (in blue, including other cities as well as rural sources) is composed of $NO_x$, PPM and $NH_3$ in different relative proportions (indicated by the arrow). The "cc" and "bg" symbols represent the city centre receptor and the background location used for the increment approach, respectively.*

Based on the formulations provided in Table 1 and equation (4), the expressions to calculate the
city and background components for the theoretical example presented above are detailed in
Table . While these formulations are relatively straightforward for potential impacts and
increments, it is more complex for the tagging method. The city tagging component is the sum of
all PM species that are directly related to the city emissions. This includes PPM and $NO_3$ that are
related to the PPM and $NO_x$ city emissions, respectively. For the background component, it
includes PPM, $NO_x$ and also $NH_4$ that is related to the $NH_3$ emissions. Tagging allows following
the $NO_x$ and $NH_3$ emitted compounds through their chemical processes and transformations until
they create $NO_3$ and $NH_4$, respectively that can be attributed to their respective sources. As $NO_x$
is emitted by both sources, the total $NO_3$ must be fractioned and attributed to each single source.




In our example, the NO₃ fraction attributed to the city depends on the ratio of the available NOₓ
precursor at the location of interest ($\beta = \frac{NOx_{city(cc)}}{NOx(cc)}$). A similar process is used to calculate the
background component.
This example is used to compared the increment (INC), tagging (TAG) and potential impact (PI)
SA approaches.

| Potential Impact | | |
|---|---|---|
| City | | $PM_{city}^{PI\alpha}(cc) = \dfrac{PM(cc) - PM_{city^{\alpha}}(cc)}{\alpha}$ |
| Background | | $PM_{bg}^{PI\alpha}(cc) = \dfrac{PM(cc) - PM_{bg^{\alpha}}(cc)}{\alpha}$ |
| **Increment** | | |
| City | | $PM_{city}^{INC}(cc) = PM(cc) - PM(bg)$ |
| Background | | $PM_{bg}^{INC}(cc) = PM(bg)$ |
| **Tagging** | | |
| City | | $PM_{city}^{TAG}(cc) = \displaystyle\sum_{E}^{city} PM_{E}(cc) = PPM_{E(PPM)_{city}}(cc) + \beta NO3^{-}{}_{E(NO2)_{city}}(cc)$ |
| Background | $PM_{bg}^{TAG}(cc) = \displaystyle\sum_{E}^{bg} PM_{E}(cc) = PPM_{E(PPM)_{bg}}(cc) + (1-\beta)NO3^{-}{}_{E(NO2)_{bg}}(cc) + NH4^{+}{}_{E(NH3)_{bg}}(cc)$ | | |

*Table A1: Formulations for the potential impacts, increments and tagging approach for the example presented in Figure . The*
*indicator for all methods and components is the total particulate matter mass (PM). The SA method is indicated as superscript*
*(PIα, INC or TAG) whereas the source (city or bg) is in subscript. The receptor is the city center (cc) while the rural location*
*selected for the increment approach is denoted by "bg". For the tagging, the source subscript is also expressed directly as*
*emissions (E) distinguishing each compound (within brackets).*
Figure  shows the city and background contributions obtained with the three SA methods,
differentiating two options for the PI one: 100% (PI100) and 20% reduction of the sources
(PI20). The figure also distinguishes four situations characterized by different background
compositions.
1. No background: When no background is present (top left), the city NOₓ emissions do not
form PM, only PPM emissions do. In such cases, all methods deliver the same response.
2. PPM background: When the background is composed of PPM only (top right), no
secondary species are formed. All methods agree with the exception of the increment
approach. This is due to the non-fulfilment of one of its underlying assumptions, i.e. the
lack of spatial homogeneity of the background which affects differently the rural and city
locations (indicated by "cc" and "bg" in Figure , respectively).





3. SEC background with $NH_3 > NO_x$: When secondary background precursors ($NO_x$ and $NH_3$) reach the city (bottom row), SA methods deliver different results because they manage differently non-linear processes. When $NH_3$ is more abundant than $NO_x$ (bottom left), the PI100 method does not preserve additivity (discussed in the "concepts" section), i.e. the sum of the two components exceeds the total PM concentration. As seen from the results and also from Table , this is not the case for the increment and tagging approaches that are constructed to be additive.

4. SEC background with $NH_3 < NO_x$: When $NH_3$ is less abundant than $NO_x$ (bottom right), differences remain important between the tagging, potential impacts and increment approaches but additivity is preserved for both PI100 and PI10 that provide identical responses.

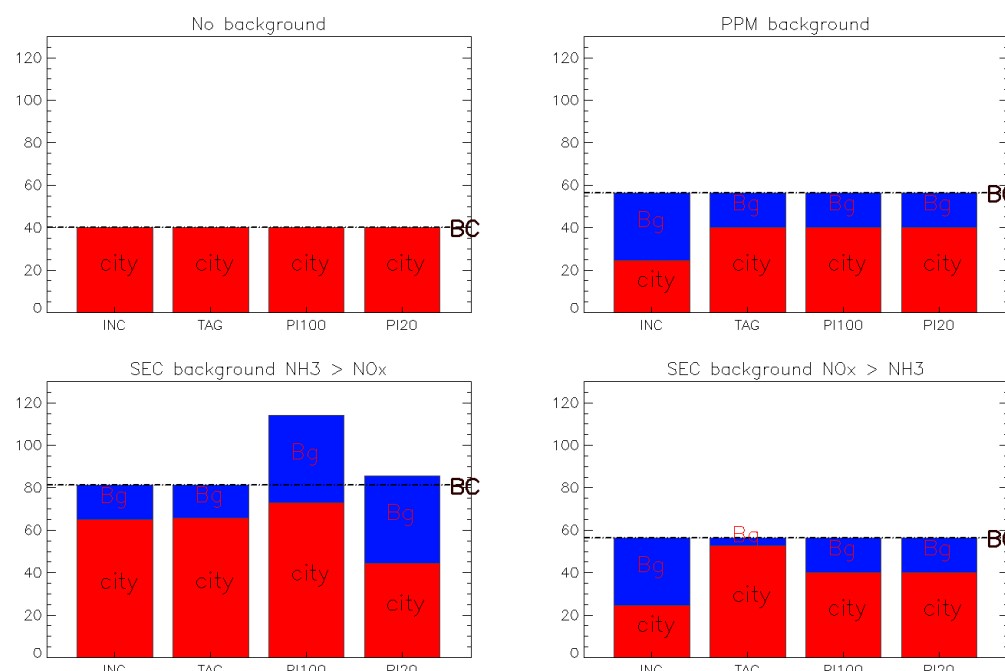

*Figure A2: Comparison of the city (red) and background (blue) components for 4 approaches applied on the theoretical examples described in Figure . Results are expressed for different types of background: (top left) no background; (top right) background limited to PPM; (bottom left) background limited to secondary but with $NH_3 > NO_x$ and (bottom right) background limited to secondary but with $NH_3 < NO_x$.*