# Peer review of "Why is the city's responsibility for its air pollution often underestimated? A focus on PM2.5"

_Atmospheric Chemistry and Physics, 2021_

## Author Response (AR1)

**Response to comment by Peter Huszar**

A very interesting paper which enlights the great influence of the method chosen for quantifying the causes of air pollution in cities. We made too similar papers dealing with these causes that I now offer into the authors' attention as potentially citable too. The first gives an answer on the question posed in this paper using the annihilation method - it shows that up to half of the pollution levels in selected European cities are caused by "nonurban" sources. The second brings light to the importance of including the urban canopy meteorological forcing in such model calculations: ignoring them significantly increases the impact of urban emissions on both local and regional air-quality.

Thanks for your positive comments. We included the two suggested references in the document. For the second, we added a paragraph in Section 3.5 (assumptions and uncertainties) to discuss the aspects related to meteorology where we refer to your work on impact of including the urban canopy in meteorological simulations. We added the following:

"On the meteorological side, the estimation of wind speed, PBL height and/or turbulence intensity will largely influence the dispersion of city emissions and uncertainties in these will therefore impact the calculation of city contributions. While the impact of meteorological parameterization on air quality has been extensively assessed from regional to urban cases (De Meij et al., 2009; (De Meij et al. 2015; De Meij et al, 2018; Jiang et al., 2020), only few studies assessed their importance on city contributions. One of these (Huszar et al. 2021) shows e.g. that the inclusion of an urban canopy meteorological forcing on multi-year simulations largely impacts the estimation of the city responsibility."

**Response to comment by Anonymous Referee #1**

The manuscript deals with an important question: who is responsible of air quality in cities? The issue is discussed from a technical point of view that tries to make order and to define common methodologies, useful to develop solid indications to policy makers and to compare different studies. It is discussed in the text, but it must be underlined that the spatial point of emission is not always the same of the responsability to take a measure (local emissions don't always mean local responsabilities).

We clearly repeated this important statement in the conclusions as follows: "This can be explained by the complex processes driving the formation of some pollutants like PM2.5, for which there is not a simple relationship between emissions and concentrations (in other words, local emissions don't always imply local responsibilities)."

From technical point of view, despite the fact that are discussed in particular two systems (SHERPA and EMEP), also the choice of many paremters (the border conditions, for example or the spatial distribution of the emission inventroy used) can affect the results.

We agree and now reformulated Section 3.5 to make clear that other parameters may influence the calculation of the city contribution as well. We also mentioned explicitly the issue of the spatial distribution when emission are discussed.

In any case the work is interesting and valueable.

**Response to comment by Anonymous Referee #2**
The manuscript sent by the authors is entitled with a remarkable question. Then, the authors present a methodological approach to address Source Appointment, proposing nomenclature to harmonize future studies. The authors then applied their methodologies to the results of models SHERPA and EMEP, to answer the question of the title. The manuscript is well written and brings to the table that air pollution is an environmental and complex problem, which requires local and regional/global solutions and ideas. However, I feel that there are some issues that need to be addressed by the authors:
Main:
The authors are too ambiguous when they are answering the question of the title. The results of the responsibility in core cities is lower than FUA regions, with values from around 20%, but higher in other cities and periods of time. Then, I think the authors could propose a threshold value to emphasize when the city is responsible for their air pollution or not, and when.

We agree with the Reviewer that we do not answer precisely enough the question raised by the title. On the other hand, we see some issues in specifying a threshold. Indeed, this threshold might depend on the pursued objective. For example, a city contribution of 2 on a total of 26 ug/m3 would be considered as limited in absolute and relative terms but could be seen as important to reach compliance with the air quality directive standards (25 ug/m3). This threshold would therefore become concentration dependent. We opted for changing the title so that it reflects better the current content of the work. The new title now reads:
 "Why is the city's responsibility for its air pollution often underestimated? A focus on PM2.5"

Can be this generalized by season? Are the cities responsible for dry and cold days? What about on under a windy pattern? Clarifying these issues in an explicit way would clarify the answer to the main question.

We agree that these are important points to clarify but we believe these go beyond the scope of this paper. We however mentioned these points more explicitly in the text and refer to an upcoming work that discusses these aspects. We added the following sentence in Section 3.4: "Correlating low and high city contributions to meteorological factors (cold vs warm days, windy vs calm situations…) is beyond the scope of this work. This point is however addressed in Pisoni et al. (2021)."

Pisoni et al. 2021. A new methodology to evaluate the effectiveness of local policies during high PM2.5 episodes: application on 10 European cities. Submitted to ACP

"emission inventories are easily seen as the scapegoat if a mismatch is found between modelled and observed concentrations of air pollutants" (Pulles and Heslinga, 2010). Between lines 481 and 487, the authors mention the uncertainty of emissions inventories, citing residential emissions and resuspension of particles, which is very good. However, the authors do not mention anything about the uncertainty of numerical modelling of meteorology. How was the wind speed? Are situations of wind speed higher than observations? This might result in lower air pollutant concentrations. Furthermore, how is generally PBL numerically represented in Europe by the meteorological models used in the EMEP/SHERPA database? Please, discuss.

We reformulated the Section 3.5 and added a paragraph to discuss meteorological aspects. This paragraph now reads as:

"On the meteorological side, the estimation of wind speed, PBL height and/or turbulence intensity will largely influence the dispersion of city emissions and uncertainties in these will therefore impact the calculation of city contributions. While the impact of meteorological parameterization on air quality has been extensively assessed from regional to urban cases (De Meij et al., 2009; (De Meij et al. 2015; De Meij et al, 2018; Jiang et al., 2020), only few studies assessed their importance on city contributions. One of these (Huszar et al. 2021) shows e.g. that the inclusion of an urban canopy meteorological forcing on multi-year simulations largely impacts the estimation of the city responsibility."

Minor:
In the abstract, lines 533 and 633 the authors mentioned "the impact of spatial averaging leads to an average factor of 2 difference in city responsible". I suggest rephrasing these sentences in a more explicitly way.

We reformulated it as follows: For the 150 EU large cities selected in our study, the different choices made for the indicator, the receptor and the source each lead to an average factor 2 difference in terms of city contribution.

Lines 256 and 590. This paragraph consists of only one sentence. Please merge with another paragraph.

Done

Lines 320-324. I was really glad that the authors mentioned particles emissions from electric vehicles

Lines 422 to 424. Why it is interesting?

We removed the beginning of the sentence.